# Pharmacist-led Medication Counseling for Patients Undergoing Hemodialysis: A Path to Better Adherence

**DOI:** 10.3390/ijerph17072399

**Published:** 2020-04-01

**Authors:** Lolwa Al-Abdelmuhsin, Maha Al-Ammari, Salmeen D Babelghaith, Syed Wajid, Yousef A Asiri, Mansour S Almetawazi, Sultan M. Alghadeer, Mohamed N. Al-Arifi

**Affiliations:** 1Pharmaceutical Care Services, Ministry of the National Guard—Health Affairs, Riyadh 1515, Saudi Arabia; alabdelmuhsinolo@ngha.med.sa (L.A.-A.); AmmariMa@NGHA.MED.SA (M.A.-A.); 2King Abdullah International Medical Research Center, Riyadh 1515, Saudi Arabia; 3King Saud bin Abdulaziz University for Health Sciences, Riyadh 11481, Saudi Arabia; 4Clinical Pharmacy Department, College of Pharmacy, King Saud University, Riyadh 11451, Saudi Arabia; sbabelghaith@ksu.edu.sa (S.D.B.); yasiri@ksu.edu.sa (Y.A.A.); mansour@ksu.edu.sa (M.S.A.); salghadeer@ksu.edu.sa (S.M.A.); malarifi@ksu.edu.sa (M.N.A.-A.)

**Keywords:** renal replacement therapy, hemodialysis, satisfaction, counselling

## Abstract

*Objective:* The primary objective was to assess the satisfaction of patients undergoing hemodialysis regarding counseling services provided by pharmacists. The secondary objectives were to compare the effect of years on dialysis and the presence of comorbidities on patient satisfaction. *Methods:* A total of 138 patients were included in the study, and all demographic and clinical variables were retrieved from the dialysis unit records of King Abdulaziz Medical City over a period of 4 months from July to October 2015. Chi-square test and Fisher’s exact test were used for group comparisons at a significance level of 0.05. *Results:* Most patients aged between 51 and 75 years and had been on dialysis for 1 to 5 years; 94.9% of them had comorbidities. The overall satisfaction of patients toward pharmacy services was excellent (77.5%), and approximately 38.4% of patients thought that pharmacists were providing clear information about their prescribed medications. In addition, 55.8% of the patients did not know that hemodialysis could affect the efficacy of their medications. *Conclusions:* Patients undergoing hemodialysis were somewhat satisfied with the counseling provided by the pharmacist. Moreover, there is a need for educational programs for patients undergoing hemodialysis that would increase awareness among hospital pharmacists to improve patients’ medication knowledge.

## 1. Introduction

Chronic kidney disease (CKD) is a progressive condition, and when renal function of the patient deteriorates to end-stage renal disease (ESRD) (glomerular filtration rate [GFR] less than 15 mL/min), CKD becomes irreversible and requires life-long renal replacement therapy (RRT) [1]. The number of patients with ESRD has been increasing steadily in incidence and prevalence over recent years. Globally, the number of patients with ESRD was estimated to be 3,200,000 at the end of 2013 with around 6% growth rate, and approximately 2,522,000 patients underwent RRT (hemodialysis or peritoneal dialysis) [2]. The number of patients with ESRD undergoing RRT in Saudi Arabia was estimated to be 3357 at the end of 1993, which increased to 16,897 (513 cases per million population) at the end of 2015, with 15,590 patients undergoing hemodialysis treatment and 1307 undergoing peritoneal dialysis [3].

In patients with CKD and comorbid disorders such as diabetes, hypertension, anemia, neuropathy, and electrolyte abnormalities, pharmacist intervention improves the outcomes of comorbidities [4,5,6,7,8]. In addition, patients with CKD usually have numerous comorbidities, and many are prescribed medications [8]. Highly prescribed-medications lead to impaired adherence to drug therapy [8]. It is known that patient counseling and providing instructions help in promoting the proper utilization of drugs, which may lead to effective therapeutic results and medication adherence [4]. Medication counseling can be defined as providing medication information orally or in a written procedure to the patients or their representatives on instructions of use, information on side effects, precautions, storage, diet, and lifestyle modifications [5]. Primary studies revealed that through medication counseling, pharmacists might recognize and solve drug-related problems, improve patient’s knowledge about the appropriate use of drugs, increase patient satisfaction with the pharmacist’s service, and consequently optimize patient quality of care [4]. Another study conducted in 2014 showed that education and counseling by clinical pharmacists in patients undergoing hemodialysis led to a clinically and statistically significant improvement in the quality of life of those patients [9].

The assessment of patient satisfaction with outpatient pharmacy services is essential for patient health. As a result, it will help to identify patients requiring improvements in the service provided [10]. Medication counseling services provided by pharmacists are essential services to improve patient adherence and quality of life [4,11]. According to the Joint International Pharmaceutical Federation and World Health Organization guidelines, good pharmacy practice is defined as the practice that responds to the needs of people who use the pharmacists’ services to provide optimal and evidence-based care [12]. The most common mistakes by pharmacists that affect the quality of counseling and patient safety include providing incorrect instructions to patients about their prescribed medication(s), duration of the counseling, and the communication skills of the pharmacist [13]. There are a couple of strategies suggested to improve the counseling service including providing written medication instruction to the patients [14], and pharmacist-managed clinics for education and counseling of patients undergoing dialysis [15]. 

The outpatient pharmacy at King Abdulaziz Medical City (KAMC) in Riyadh has a dedicated window to receive/counsel patients undergoing dialysis about prescribed medications. They offer to deliver medication to the dialysis unit if the patient cannot go to the pharmacy. Moreover, KAMC has a discharge counseling program that dedicates a pharmacist to counsel each patient who underwent dialysis in the dialysis unit before discharge. Discharge counseling pharmacists play a major role in patient counseling by reviewing discharge medications, enabling them to identify errors if present, documenting all medication-discharge-counseling statistics in the pharmacy counseling system, and reporting any medication errors or near misses. 

A prospective, pre–post study conducted in the hemodialysis unit of KIST Medical College and Teaching Hospital, Lalitpur, Nepal. This study is similar to ours in terms of the targeted population, as well as the assessment of pharmacist counseling, and patient satisfaction regarding the service provided but differs regarding objectives (i.e., to measure patient knowledge, attitude, and practice outcomes). The researchers concluded that pharmacist-provided counseling is effective in improving the knowledge, attitude, and practice of patients toward disease management [16]. In contrast, we found additional related studies in terms of targeted population and the assessment of the satisfaction of patients undergoing hemodialysis but with other objectives. For example, a survey study was undertaken by the American Association of Kidney Patients on 977 patients with ESRD found that patients were moderately satisfied with their pre-treatment education [17]. Another cross-sectional survey concluded that patients undergoing hemodialysis are least satisfied with the complex aspects of care [9].

To the best of our knowledge, no studies have been conducted in Saudi Arabia to assess the satisfaction of patients undergoing hemodialysis with counseling services provided by pharmacy departments. To fill this gap in the literature, we conducted this study to assess the satisfaction of patients undergoing hemodialysis regarding the counseling services provided by the pharmacists as a primary objective. The secondary objectives were to compare the effect of years on dialysis and the presence of comorbidities on patient satisfaction. 

## 2. Methods

This was a cross-sectional survey that was distributed among patients undergoing hemodialysis in the KAMC-Central Region (KAMC-CR). The institutional review board of the King Abdulla Medical Research Center approved the study in August 2015.

### 2.1. Eligibility Criteria 

Patients were enrolled in the study for the final analysis if they were 18 years old or more, on hemodialysis for at least 3 months, and were eligible at KAMC-CR. Patients were excluded if they were on peritoneal dialysis, had cognitive impairment, had been diagnosed with dementia, or had confirmed Middle East Respiratory Syndrome Corona Virus. A total of 224 patients undergoing hemodialysis were screened, of which 86 patients were excluded and 138 were included for final analysis.

### 2.2. Outcomes 

The primary outcome of this study was to assess the satisfaction of patients undergoing hemodialysis regarding the counseling services provided by pharmacists. 

The secondary outcomes were to compare the effect of years on dialysis and the presence of comorbidities on patient satisfaction.

### 2.3. Instruments 

A face-to-face patient satisfaction survey that contained five parts was administered to patients undergoing hemodialysis (Appendix A). The cover sheet included a brief introduction to the study and patient serial number. The first part of the survey recorded demographic data and hemodialysis duration. The second part detailed medication claiming from the pharmacy (pharmacy location and person responsible for collecting the medication), preferable counseling time (before, during, or after a dialysis session), and preferable health care provider to conduct the counseling session (physician, pharmacist, or nurse). The third part assessed the overall rating of outpatient pharmacy services by patients undergoing hemodialysis. The fourth part related to evaluating the status of pharmacy counseling services.

The fifth part requested the opinions of patients undergoing hemodialysis on improving the counseling services by providing a hotline for medication counseling and/or providing educational material (e.g., a booklet), their knowledge regarding the effect of a dialysis session on the efficacy of their prescribed medication, and their willingness to know more about their medications. The last page of the survey asked for consent, requiring the official name and signature of the patient. A pilot study was conducted among five respondents to test the reliability of the questionnaire. The pilot study respondents were patients undergoing hemodialysis, were not included in the results of the study, and were not contacted. Cronbach’s alpha was 0.7.

#### Data Analysis 

Collected data were combined in an Excel sheet (2013) that was exported to SPSS sheet (IBM SPSS^®^ version 22, (SPSS Inc., Chicago, IL, USA) for statistical analysis. 

Results of statistical analysis for descriptive data were summarized as numbers and percentages. This study used the chi-square test and Fisher’s exact test for all analyses, and a *p* value < 0.05 was considered statistically significant. 

## 3. Result

### 3.1. Patient Information

A total of 224 patients undergoing hemodialysis were screened, 86 patients were excluded, and only 138 patients were included in the final analysis. Overall, 50.7% were male, 51.4% were aged between 51 and 75 years and approximately 49.3% of patients underwent dialysis for 1 to 5 years (Table 1). 

### 3.2. Patient Preference and Opinions

A total of 48.6% of the study sample requested relatives to claim prescribed medications from the pharmacy, and 71% of them received medications from an ambulatory care pharmacy. Approximately 42.8% of patients undergoing hemodialysis preferred to be counseled about their medications during their dialysis session. Almost 37.7% of the patients thought that the physician should be the one to do the medication counseling (Table 2).

There was no statistically significant difference between patient preferences and thoughts versus years on dialysis. However, the difference between years on dialysis and the percentage of patients who prefer to receive medication through the nurse was statistically significant (*p* value 0.016). This result could have been affected by the higher number of patients who have been on dialysis for more than 1 year than the number of patients who have been on dialysis for less than 1 year, as well as by the number of patients who do not know about this service (Table 3).

The idea of providing educational medication material to patients undergoing hemodialysis containing all medication details was agreed upon by 77.5% of the patients. Although 81.2% of patients think that providing a hotline for medication counseling would improve compliance to the medication, a total of 55.8% of the patients did not know that hemodialysis could affect the efficacy of their medication(s): 64.5% of them showed an interest in knowing more about their prescribed medications (Table 4).

### 3.3. Overall Patient Satisfaction

Most patients undergoing hemodialysis (77.5%) rated the pharmacy services as “excellent” and 73.9% gave an “excellent” rating to pharmacy medication counseling services (Table 5). There was no statistically significant difference in overall patient satisfaction versus years on dialysis (Table 6). 

### 3.4. Pharmacist Counseling Behavior

Approximately 65% to 67% of patients were not eligible to answer this part of the survey because they did not claim their own medication from the pharmacy. Approximately 38.4% of patients undergoing hemodialysis thought that pharmacists were providing clear information about their prescribed medications. In contrast, 19.6% of the patients thought that the pharmacist did not encourage them to ask questions regarding their medications (Table 7).

## 4. Discussion

This study assessed the opinions of patients undergoing hemodialysis regarding medication counseling. The current study reported that 77.5% of patients undergoing hemodialysis agreed to receiving booklets containing information about their medication. A study was conducted to assess the efficacy of medication counseling and the use of a drug booklet on the level of treatment compliance determined by improved levels of glycated hemoglobin (HbA1C) and on medication adherence in patients with type 2 diabetes mellitus. This study found that both HbA1C and medication adherence improved after the intervention [18]. 

In this study, most patients undergoing hemodialysis (81.2%) believed that providing a hotline for medication counseling would improve compliance to their medication. This finding supports the efficacy of hotline research determined in previous studies. A qualitative, exploratory study was conducted to study the perceptions of a group of culturally and linguistically diverse participants, with comorbidities such as CKD, to determine factors that affected their medication self-efficacy with the use of motivational interviewing using the telephone [19]. This study found that poor knowledge about drugs delayed the confidence necessary for optimal disease self-management [19].

Renal deficiency decreases the clearance of some medications [20]. Thus, the drug dose should be adjusted for patients with acute or CKD [12]. In this study, 44.2% of patients undergoing hemodialysis knew that hemodialysis affects the efficacy of their medications. Because this study used a different questionnaire, our results cannot not be directly compared with those of previous studies. Furthermore, a study was conducted in India to assess the knowledge of patients undergoing hemodialysis about their medications and the impact of education programs on their medication knowledge. The findings of this study reported that patients undergoing hemodialysis had very poor knowledge of the name, use, and dose of their medications [21]. Thus, insufficient knowledge of patients regarding their medications is a contributing factor in medication non-adherence.

This study evaluated the satisfaction of patients with pharmacist services. Improving the level of pharmaceutical care can be achieved by utilizing different approaches to improve patient satisfaction [15]. This is because hemodialysis is multifaceted and requires a multi-disciplinary approach. Therefore, the evaluation of patient satisfaction is vital to assessing healthcare outcomes [10,15]. The results of this study reveal that most patients undergoing hemodialysis (77.5%) were satisfied with the pharmacist services. In addition, about 74% of patients undergoing hemodialysis gave an “excellent” rating to pharmacy medication counseling services. We compared our findings with those of a study conducted in central Saudi Arabia to assess the influence of counseling on patients undergoing hemodialysis and their quality of life. It found that most patients undergoing hemodialysis were satisfied with healthcare provider services (88.7%), and 88% of patients undergoing hemodialysis received counseling from their healthcare providers (physicians, pharmacists, and nurses) on medication use [22]. This could be related to the counseling of different healthcare providers. In addition, this study reported that 50% of patients undergoing hemodialysis rated services provided by healthcare providers as “excellent” [22]. 

The level of pharmacist care provided to patients undergoing hemodialysis at a tertiary hospital in Riyadh was reported by most patients as “excellent” and “very good.” However, there were some patients who stated that the services provided by pharmacists were poor. This is a concern and it would therefore be important to know the cause behind such a rating by the patient.

Finally, education programs for pharmacists and patients undergoing hemodialysis should be applied to improve the understanding, detection, and management of CKD. A study was conducted to determine the impact of a pharmacist intervention on the outcome of renal function in patients with CKD. The results of this study showed that the renal profile improved after the intervention of pharmacists compared with that in the control group [23]. Another survey was conducted to assess the impact of clinical pharmacist counseling on the knowledge, attitude, and practice of patients with CKD (*n* = 64). The study revealed that the mean knowledge, attitude, and practice scores for disease management were improved after the intervention (*p* < 0.05) [16].

## 5. Limitations

This study has some limitations. Pharmacists were from a single hospital in central Saudi Arabia. So, the pharmacists were not representative of the entire population of hospital pharmacists in Saudi Arabia as a whole. In addition, this study did not assess in detail the satisfaction of hemodialysis patients about the role of pharmacists.

## 6. Conclusions

In general, patients undergoing hemodialysis were found to be satisfied with the counseling provided by the pharmacist. These results offer future reference for the implementation of the counseling service within hospital pharmacy practices. Finally, more studies with a multicenter focus on patient satisfaction with pharmacy services in Saudi Arabia are needed.

## Figures and Tables

**Table 1 ijerph-17-02399-t001:** Characteristics of hemodialysis patients.

Gender	Number (*n*)	Percentage (%)
Male Female	70 68	50.7 49.3
Age		
18–30 years 30–50 years 51–75 years 75+ years	20 30 71 17	14.5 21.7 51.4 12.3
Language		
Arabic only English only Arabic & English	132 0 5	96.4 0 3.6
Education level		
Primary school or less Intermediate school High school Bachelors Advanced degree	92 14 20 11 1	66.7 10.1 14.5 8 0.7
Occupational status		
Employed Unemployed	17 121	12.3 87.7
Marital status		
Single Married Divorced Widowed	29 85 4 20	21 61.6 2.9 14.5
Number of children		
None Five or less Ten or less More than ten	29 43 47 19	21 31.2 34.1 13.8
Duration of dialysis		
3–6 months 6–12 months 1–5 years 5+ years	3 9 68 58	2.2 6.5 49.3 42
Comorbidities		
No comorbidities comorbidities	7 138	5.1 94.9

**Table 2 ijerph-17-02399-t002:** Patient preference.

Statements	Number (*n*)	Percentage (%)
**Who gets your medication usually from the pharmacy**		
Him/herself Relatives House maid or driver Through the nurse.	54 67 17 57	39.1 48.6 12.3 41.3
**I usually claim my medications from**		
Ambulatory Care pharmacy Emergency Pharmacy Both	98 8 32	71 5.8 23.2
**The best time for pharmacist counseling during the day of hemodialysis session is**		
Before dialysis session During dialysis session After dialysis session Do not know	9 59 21 49	6.5 42.8 15.2 35.5
**Who is supposed to give you the proper information about your prescribed medication**		
Physician Pharmacist Nurse Do not know	52 44 8 34	37.7 31.9 5.8 24.6

**Table 3 ijerph-17-02399-t003:** Patient preference and thoughts vs. years on dialysis.

Statements	3–6 Months	>6–12 Months	>1–5 Years	>5 Years	*p*-Value
	Number (Percentage %)				
**Who gets your medication usually from the pharmacy**					
Him/herself Relatives House maid or driver Through the nurse.	0 (0) 3 (2.2) 0 (0) 0 (0)	2 (1.4) 4 (2.9) 2 (1.4) 2 (1.4)	28 (20.3) 35 (25.4) 9 (6.5) 23 (16.7)	24 (17.4) 25 (18.1) 6 (4.3) 32 (32.2)	0.370 0.249 0.750 0.016
**I usually claim my medications from**					
Ambulatory Care pharmacy Emergency Pharmacy Both	2 (66.7) 1 (33.3) 0(0)	5 (55.6) 0 (0) 4 (44.4)	47 (69.1) 4 (5.9) 17 (25)	44 (75.9) 3 (5.2) 11 (8)	0.226
**The best time for pharmacist counseling during the day of hemodialysis session is**					
Before dialysis session During dialysis session After dialysis session Do not know	0 (0) 2 (66.7) 1 (33.3) 0 (0)	1 (11.1) 4 (44.4) 1 (11.1) 3 (33.3)	4 (5.9) 29 (42.6) 13 (19.1) 22 (32.4)	4 (6.9) 24 (41.4) 6 (10.3) 24 (41.4)	0.833
**Who is supposed to give you the proper information about your prescribed medication**					
Physician Pharmacist Nurse Do not know	2 (66.7) 0 (0) 0 (0) 1 (33.3)	3 (33.3) 5 (55.6) 0 (0) 1 (11.1)	26 (38.2) 22 (32.4) 3 (4.4) 17 (25)	21 (36.2) 17 (38.6) 5 (8.6) 34 (24.6)	0.757

**Table 4 ijerph-17-02399-t004:** Opinion of patients toward statements.

Statements	Agree	Disagree
	Number (Percentage %)	
**I prefer to have a booklet about my medication**	107(77.5)	31(22.5)
**I think providing a hotline for medication counseling will improve my compliance to the medication**	112(81.2)	26(18.8)
**I know that hemodialysis will affect the efficiency of my medications**	61(44.2)	77(55.8)
**I am interested to know more about my medication**	89(64.5)	49(35.5)

**Table 5 ijerph-17-02399-t005:** Overall patient satisfaction.

Statements	Poor	Fair	Good	Very Good	Excellent
	Number (Percentage %)				
**The overall pharmacy services for dialysis-dependent patient regarding their prescribed medication**	5 (3.6)	2 (1.4)	7 (5)	17 (12.3)	107 (77.5)
**The pharmacist counseling regarding your prescribed medication**	16 (11.6)	1 (0.7)	10 (7.2)	9 (6.5)	102 (73.9)

**Table 6 ijerph-17-02399-t006:** Overall, patient satisfaction vs. years on dialysis.

Statements	3–6 Months	>6–12 Months	>1–5 Years	>5 Years	*p*-Value
	Number (Percentage %)				
**The overall pharmacy services for dialysis-dependent patient regarding their prescribed medication**					
Poor Fair Good Very good Excellent	0 (0) 0 (0) 1 (33.3) 0 (0) 2 (66.7)	1 (11.1) 0 (0) 0 (0) 1 (11.1) 7 (77.8)	1 (1.5) 2 (2.9) 4 (5.9) 8 (11.7) 53 (78)	3 (5.1) 0 (0) 2 (3.4) 8 (13.8) 45 (77.6)	0.664
**The pharmacist counseling regarding your prescribed medication**					
Poor Fair Good Very good Excellent	0 (0) 0 (0) 1 (33.3) 0 (0) 2 (66.7)	0 (0) 0 (0) 3 (33.3) 0 (0) 6 (66.7)	7 (10.3) 0 (0) 2 (2.9) 4 (5.9) 55 (80.8)	9 (15.5) 1 (1.7) 4 (6.9) 5 (8.6) 39 (67.2)	0.219

**Table 7 ijerph-17-02399-t007:** Behaviors of the pharmacist counseling patients.

Statements	Usually	Sometimes	Never	N/A
	Number (Percentage %)			
The pharmacist encourages me to ask any questions regarding my prescribed medication	24 (17.4)	21 (15.2)	27 (19.6)	66 (47.8)
The pharmacist who dispenses your prescribed medication listens carefully to you	36 (26.1)	26 (18.8)	9 (6.5)	67 (48.6)
The pharmacist who dispenses your prescribed medication explains things in a way you can understand	53 (38.4)	13 (9.4)	7 (5.1)	65 (47.1)
The pharmacist who dispenses your prescribed medication gives you enough time for counseling	37 (26.8)	22 (15.9)	14 (10.1)	65 (47.1)

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
