# Peer review of "Pharmacist-led Medication Counseling for Patients Undergoing Hemodialysis: A Path to Better Adherence"

_ijerph, 2020, doi:10.3390/ijerph17072399_

Round 1
Reviewer 1 Report
The study presented by Al_Abdelmushin et al deserves some merit based on its attempt to study the level of patients satisfaction with pharmacy led counseling during hemodialysis treatments and how the level of satisfaction may be an opportunity to improve outcomes. However, I have some major concerns with regard to the manuscript.
The English proficiency, while acceptable, was not "great" and took away from the focus of the manuscript.
The title is very wordy and not specific. As I understand the paper it would be good to go with a simpler title such as: "Pharmacist led Medication Counseling in Patients on Hemodialysis- A Path to Better Adherence?"
The abstract and introduction confuse me. The first sentence of the second paragraph makes a statement about satisfaction with pharmacy services but gives no references. Why do the authors believe this? This reader is unaware of what role pharmacists fill in Saudi dialysis facilities? Do pharmacists play an essential role? How would they help ESRD outcomes? Would they help with adherence of Phos binders or vitamin D?
Of course patient satisfaction is an important outcome to measure in a population that is overwhelmed with the burden of their disease and as a result likely non-adherent to their complex dialysis and medication regimens. Are the authors trying to gauge an improvement in adherence by providing pharmacy services? Why are they examining who picks up their medication. How would that impact on their adherence? Is patient satisfaction all that they are interested in?
methods:
Did the patients consent to participate?
one of the exclusion criteria is being dependent on someone to get meds from pharmacy. But then, the survey asks about who gets their medication from pharmacy and having family members get meds from pharmacy is a major component of the results
why were the 86 patients excluded?
were surveys given to the subjects while they were on dialysis?
the instrument was validated using 5 subjects.....how was it validated.
I don't understand the use of Cronbachs alpha to test the reliability of the questionnaire. This statistical technique is used to measure the applicability of the total score on a questionnaire to the score on the different categories of its components; or the applicability of the mean test score, to the variability of individuals taking it. Is this what the authors are doing?
Results:
In order for the results to be less confusing, the authors must provide in the methods the exact pharmacist intervention and the outcomes they were expecting. In its current form it is very hard to follow.
one of the strengths of this study is that the majority of the patients had a limited education such that this pharmacy counseling would be extra beneficial. Please highlight this strength.
Discussion
The first paragraph needs work. Would suggest reintroducing the hypothesis here and following up with what the study showed, and how it supported or refuted the hypothesis.
What do the authors mean by patient dialysis information? How can a pharmacist help with that?
Author Response
Response to Reviewer 1 Comments
The English proficiency, while acceptable, was not "great" and took away from the focus of the manuscript.
The English editing was done and English certificate will be attached
The title is very wordy and not specific. As I understand the paper it would be good to go with a simpler title such as: "Pharmacist led Medication Counseling in Patients on Hemodialysis- A Path to Better Adherence?"
Totally agree, this title not fully understandable. We changed to the following “Pharmacist led Medication Counseling in Patients on Hemodialysis- A Path to Better Adherence”. Thank you very much
The abstract and introduction confuse me.
- The first sentence of the second paragraph makes a statement about satisfaction with pharmacy services but gives no references. Why do the authors believe this? The references was added and cited as Abdrrahman S. Surur, Fitsum S. Teni, Genet Girmay, et al. Satisfaction of clients with the services of an outpatient pharmacy at a university hospital in northwestern Ethiopia: a cross-sectional study. BMC Health Services Research (2015) 15:229.
- This reader is unaware of what role pharmacists fill in Saudi dialysis facilities? Do pharmacists play an essential role? How would they help ESRD outcomes? Would they help with adherence of Phos binders or vitamin D?
Totally agree, and we added the role of pharmacists at King Abdulaziz Medical City
The outpatient pharmacy at King Abdulaziz Medical City in Riyadh dedicate a window to receive/ counsel dialysis patients about all their prescribed medication. They even offer to deliver their medication to the dialysis unit, if they cannot come to the pharmacy. Moreover, King Abdulaziz Medical City in Riyadh has discharge counselling program which dedicate a pharmacist to counsel dialysis patient in the dialysis unit before discharge. Discharge counseling pharmacists play a major role in patient counseling by reviewing discharge medications, enabling them to discover errors if present, document all medication discharge-counseling statistics in the "Pharmacy Counselling System", and report any medication errors or near misses.
Of course patient satisfaction is an important outcome to measure in a population that is overwhelmed with the burden of their disease and as a result likely non-adherent to their complex dialysis and medication regimens. Are the authors trying to gauge an improvement in adherence by providing pharmacy services? Why are they examining who picks up their medication. How would that impact on their adherence? Is patient satisfaction all that they are interested in?
methods:
- Did the patients consent to participate?
Yes.
- one of the exclusion criteria is being dependent on someone to get meds from pharmacy. But then, the survey asks about who gets their medication from pharmacy and having family members get meds from pharmacy is a major component of the results
- Why were the 86 patients excluded? And we explained in methods
- Were surveys given to the subjects while they were on dialysis?
Yes. The survey done while they were on dialysis.
one of the exclusion criteria is being dependent on someone to get meds from pharmacy. But then, the survey asks about who gets their medication from pharmacy and having family members get meds from pharmacy is a major component of the results
the instrument was validated using 5 subjects. how was it validated.
I don't understand the use of Cronbachs alpha to test the reliability of the questionnaire. This statistical technique is used to measure the applicability of the total score on a questionnaire to the score on the different categories of its components; or the applicability of the mean test score, to the variability of individuals taking it. Is this what the authors are doing?
Yes :
The pilot study was done among 5 respondents to test the reliability of the questionnaire. The respondents elaborate in the pilot study were hemodialysis patients and not included in the final research or not contacted with the respondents of the study. The Cronbach’s alpha was 0.7.
Results:
In order for the results to be less confusing, the authors must provide in the methods the exact pharmacist intervention and the outcomes they were expecting. In its current form it is very hard to follow.
Answer: the corrections was made see methods section
one of the strengths of this study is that the majority of the patients had a limited education such that this pharmacy counseling would be extra beneficial. Please highlight this strength.
This study did not find any significant difference between education levels and overall patient satisfaction.
Discussion
The first paragraph needs work. Would suggest reintroducing the hypothesis here and following up with what the study showed, and how it supported or refuted the hypothesis.
Totally agree, The first paragraph was removed and started by the aims of studies and discussed the results
What do the authors mean by patient dialysis information? How can a pharmacist help with that?
It is removed because no need to it we just focus on the discussion of outcomes
Reviewer 2 Report
My response will be a general one. I have not reviewed the tables for their statistical accuracy.
I believe that this is an important study. However, the English is at times difficult to understand, making the point of some statements unclear. A bigger problem is that mention is made both in the abstract and the conclusion of the possibility of responses being affected by "gratitude to the government," even though service was found to be excellent there may be need for improvement as a result. This is a conclusion that requires explanation. There is nothing in the information presented in the body of the paper that gives reason for this conclusion. If the authors feel that they did not get an accurate result in their study because of such gratitude and the information provided by the study is both wanted and needed by pharmacists in understanding how they can improve their guidance to patients, then evidence for the importance of this gratitude in affecting the outcome of the study must be provided.
I have made a number of corrections and suggested edits directly to the document (with the tables removed). See the attached document.

Author Response
Response to Reviewer 2
Comments and Suggestions for Authors
My response will be a general one. I have not reviewed the tables for their statistical accuracy.
I believe that this is an important study. However, the English is at times difficult to understand, making the point of some statements unclear. A bigger problem is that mention is made both in the abstract and the conclusion of the possibility of responses being affected by "gratitude to the government," even though service was found to be excellent there may be need for improvement as a result. This is a conclusion that requires explanation. There is nothing in the information presented in the body of the paper that gives reason for this conclusion. If the authors feel that they did not get an accurate result in their study because of such gratitude and the information provided by the study is both wanted and needed by pharmacists in understanding how they can improve their guidance to patients, then evidence for the importance of this gratitude in affecting the outcome of the study must be provided.
Totally agree with your comments than you very much “ The results in abstract was changed as well as conclusion
The overall satisfaction of the hemodialysis patients toward the pharmacy services were excellent (77.5 %), around 38.4 % of hemodialysis patients think that pharmacists are providing understandable information about their prescribed medications. In addition, a total of 55.8 % of the patients do not know the hemodialysis could affect the efficacy of their medication. Conclusions: The hemodialysis patient had somewhat satisfactions about the counseling provided by pharmacist. In fact, there is needed for educations programs for hemodialysis patients to increase their awareness among hospital pharmacists to improve their medications knowledge.
I have made a number of corrections and suggested edits directly to the document (with the tables removed). See the attached document.
Thank you very much for great work
the manuscript requires general linguistic editing. Much of the article is difficult to understand. The reasonableness and purposefulness of the research should be described more in the introduction. The inclusion and exclusion criteria for the study should be clarified. Similarly, the description of statistical methods needs to be clarified. The tasks and role of a pharmacist in the care of a haemodialysis patient should be discussed in more detail. What is the role of the hospital pharmacist? What is the role of the pharmacist in community pharmacy? What are the conclusions of the study? What are their clinical implications? What should be further investigated? Finally, what are the limitations of the study?
inclusion and exclusion criteria was clarified.
statistical methods was clarified as “Results of statistical analysis for descriptive data were summarized as numbers and percentages. This study used the chi-square test and Fisher’s exact test for all analyses, and a P value < 0.05 was considered statistically significant. .
The tasks and role of a pharmacist in the care of a haemodialysis patient should be discussed in more detail.
We discussed the role of pharmacists in the care of a haemodialysis patient . please see the introduction section and statements were highlighted :
In patients with CKD and comorbid disorders such as diabetes, hypertension, anemia, neuropathy, and electrolyte abnormalities, pharmacist intervention improves the outcomes of comorbidities [4-8]. In addition, patients with CKD usually have numerous comorbidities, and many are prescribed medications [8]. Highly prescribed-medications lead to impaired adherence to drug therapy [8]. It is known that patient counseling and providing instructions help in promoting proper utilization of drugs, which may lead to effective therapeutic results and medication adherence [4]. Medication counseling can be defined as providing medication information orally or in written procedure to the patients or their representatives on instructions of use, information on side effects, precautions, storage, diet, and lifestyle modifications [5]. Primary studies revealed that through medication counseling, pharmacists might recognize and solve drug-related problems, improve patient’s knowledge about the appropriate use of drugs, increase patient satisfaction with the pharmacist’s service, and consequently optimize patient quality of care [4]. Another study conducted in 2014 showed that education and counseling by clinical pharmacists in patients undergoing hemodialysis led to clinically and statistically significant improvement in the quality of life of those patients [9].
The outpatient pharmacy at King Abdulaziz Medical City (KAMC) in Riyadh has a dedicated window to receive/counsel patients undergoing dialysis about prescribed medications. They offer delivery of medication to the dialysis unit if the patient cannot go to the pharmacy. Moreover, KAMC has a discharge counseling program that dedicates a pharmacist to counsel patients who underwent dialysis in the dialysis unit before discharge. Discharge counseling pharmacists play a major role in patient counseling by reviewing discharge medications, enabling them to identify errors if present, documenting all medication-discharge-counseling statistics in the pharmacy counseling system, and reporting any medication errors or near misses.
Reviewer 3 Report
The manuscript requires general linguistic editing. Much of the article is difficult to understand. The reasonableness and purposefulness of the research should be described more in the introduction. The inclusion and exclusion criteria for the study should be clarified. Similarly, the description of statistical methods needs to be clarified. The tasks and role of a pharmacist in the care of a haemodialysis patient should be discussed in more detail. What is the role of the hospital pharmacist? What is the role of the pharmacist in community pharmacy? What are the conclusions of the study? What are their clinical implications? What should be further investigated? Finally, what are the limitations of the study?
Author Response
Response to reviewer 3
The manuscript requires general linguistic editing. Much of the article is difficult to understand.
The English editing was done and English certificate will be attached
The reasonableness and purposefulness of the research should be described more in the introduction.
Thank you. totally agree, and a statement was added into introduction and highlighted
The inclusion and exclusion criteria for the study should be clarified.
It was clarified please see the methods sections
Eligibility criteria
Patients were enrolled in the study for the final analysis if they were 18 years old or more, on hemodialysis for at least 3 months, and were eligible at KAMC-CR. Patients were excluded if they were on peritoneal dialysis, had cognitive impairment, had been diagnosed with dementia, were dependent on caregivers for claiming their medications from the pharmacy, or had confirmed Middle East Respiratory Syndrome Corona Virus. A total of 224 patients undergoing hemodialysis were screened, of which 86 patients were excluded and 138 were included for final analysis.
Similarly, the description of statistical methods needs to be clarified. It was corrected as following :
Data analysis
Collected data were combined in an Excel sheet (2013) that was exported to SPSS sheet (IBM SPSS® version 22) for statistical analysis.
Results of statistical analysis for descriptive data were summarized as numbers and percentages. This study used the chi-square test and Fisher’s exact test for all analyses, and a P value < 0.05 was considered statistically significant.
The tasks and role of a pharmacist in the care of a haemodialysis patient should be discussed in more detail.
It was discussed in more detail in introduction section and highlighted
What is the role of the hospital pharmacist? What is the role of the pharmacist in community pharmacy? What are the conclusions of the study? What are their clinical implications? Finally, what are the limitations of the study?
All theses comments were corrected and added
What should be further investig discussed in more detail.ated?
Finally, education programs for pharmacists and patients undergoing hemodialysis should be applied to improve the understanding, detection, and management of CKD. A study was conducted to determine the impact of a pharmacist intervention on the outcome of renal function in patients with CKD. The results of this study showed that the renal profile improved after the intervention of pharmacists compared with that in the control group [24]. Another survey was conducted to assess the impact of clinical pharmacist counseling on knowledge, attitude, and practice of patients with CKD (n=64). The study revealed that the mean knowledge, attitude, and practice scores for disease management were improved after the intervention (p < 0.05) [16].
What are the conclusions of the study?
In general, patients undergoing hemodialysis were found to be satisfied with the counseling provided by the pharmacist. These results offer provision for the implementation of the service within hospital pharmacy practices
Round 2
Reviewer 1 Report
The authors did a good job in addressing some of the concerns that were raised on last version.
1) please address the following: if one of the exclusion criteria was: "were dependent on caregivers for claiming their medications from the pharmacy", then how can they include those subjects who: "48.6 % of the study sample requested relatives to claim prescribed medications from the pharmacy".
2) the introduction is very wordy, please avoid the detail with which you describe other studies (page 2, last paragraph). Perhaps a brief summary of the findings of these studies is sufficient.
3) do the authors provide a copy of the questionnaire in their supplemental material?
Author Response
1) please address the following: if one of the exclusion criteria was: "were dependent on caregivers for claiming their medications from the pharmacy", then how can they include those subjects who: "48.6 % of the study sample requested relatives to claim prescribed medications from the pharmacy".
2) the introduction is very wordy, please avoid the detail with which you describe other studies (page 2, last paragraph). Perhaps a brief summary of the findings of these studies is sufficient.
Totally agree, thank you, we remove some details please see the introduction section
Totally agree, so we remove that exclusion of the eligible criteria
3) do the authors provide a copy of the questionnaire in their supplemental material?
It was attached

Reviewer 2 Report
The new title,
Pharmacist-led Medication Counseling for Patients Undergoing Hemodialysis: A Path to Better Adherence
is a better description of the study.
The writing is much improved over the last version. I have a few additional suggestions for changes and a number of questions that I provide in the attached Word document as comments. One thing in particular I find confusing is an observation that is in contrast to what was claimed in the previous version (the new version states, "those with comorbidities were less satisfied with the counseling process because they had poorer understanding of their medications and wanted to learn more about them").
The conclusion has become a little weak. Please expand on it.
As well, the paper would be improved with a limitations section.
I look forward to this submission soon being ready for publication.

Author Response
The new title,
Pharmacist-led Medication Counseling for Patients Undergoing Hemodialysis: A Path to Better Adherence
is a better description of the study. Thank you very much.
The writing is much improved over the last version. I have a few additional suggestions for changes and a number of questions that I provide in the attached Word document as comments. One thing in particular I find confusing is an observation that is in contrast to what was claimed in the previous version (the new version states, "those with comorbidities were less satisfied with the counseling process because they had poorer understanding of their medications and wanted to learn more about them").
The statement was removed “” those with comorbidities were less satisfied with the counseling process because they had poorer understanding of their medications and wanted to learn more about them").
The conclusion has become a little weak. Please expand on it.
As well, the paper would be improved with a limitations section.
Limitations: This study has some limitations. Pharmacists were from a single hospital of central
Saudi Arabia. So, not representative of the entire population of hospital pharmacist’s in
Saudi Arabia as whole. we did not assess in detail the satisfaction of hemodialysis patients about the role of pharmacists.
Reviewer 3 Report
Ready for publication
Author Response
thank you very much